# Best Possible Medication History Collection by Clinical Pharmacist in a Preoperative Setting: An Observational Prospective Study

**DOI:** 10.3390/pharmacy11050142

**Published:** 2023-09-08

**Authors:** Daniele Mengato, Lisa Pivato, Lorenzo Codato, Fernanda Fabiola Faccioli, Laura Camuffo, Maria Cecilia Giron, Francesca Venturini

**Affiliations:** 1Hospital Pharmacy Department, Padova University Hospital (Azienda Ospedale-Università Padova), Via Giustiniani 2, 35128 Padua, Italy; lisa.pivato@aopd.veneto.it (L.P.); fabiola.faccioli@gmail.com (F.F.F.); laura.camuffo@aopd.veneto.it (L.C.); francesca.venturini@aopd.veneto.it (F.V.); 2Department of Pharmaceutical and Pharmacological Sciences, University of Padova, Pharmacology Building, Via Marzolo 5, 35131 Padova, Italy; lorenzo.codato@gmail.com (L.C.); cecilia.giron@unipd.it (M.C.G.)

**Keywords:** clinical pharmacist, pharmacist intervention, perioperative, surgical setting, potentially inappropriate medication, best possible medication history

## Abstract

Background: A Best Possible Medication History (BPMH) collected by clinical pharmacists is crucial for effective medication review, but, in Italy, it is often left to the nursing staff. This study aims to compare the quality and accuracy of a clinical pharmacist-documented BPMH with the current standard practice of ward staff-collected BPMH in an Italian preoperative surgical setting. Methods: A 20-week prospective observational non-profit study was conducted in a major university hospital. The study comprised three phases: a feasibility, an observational, and an interventional phase. During the feasibility phase, 10 items for obtaining a correct BPMH were identified. The control group consisted of retrospectively analyzed BPMHs collected by the ward staff during the observational phase, while interventions included BPMHs collected by the clinical pharmacist during the third phase. Omissions between the two groups were compared. Results: 14 (2.0%) omissions were found in the intervention group, compared with 400 (57.4%) found in the controls (*p* < 0.05); data collection was more complete when collected by pharmacists compared to the current modality (98.0% of completed information for the intervention versus 42.6%; *p* < 0.05). Conclusions: The involvement of a pharmacist significantly reduced the number of omissions in preoperative surgical-collected BPMHs. This intervention holds the potential to decrease the risk of medication errors associated with inaccurate or incomplete BPMHs prior to surgical hospitalization.

## 1. Introduction

Medication review is defined as a clinical intervention, frequently performed by pharmacists, aimed at improving medication safety and health outcomes by ensuring optimal medication use [1]. It consists of two consecutive phases: the identification of the best possible medication history (BPMH) and the medication reconciliation. In particular, the BPMH allows collection of complete and accurate information about the patient and his/her medication. In addition, the reconciliation examines the patient’s clinical conditions during therapeutic transitions [2]. BPMH refers to a comprehensive and accurate record of a patient’s past and current medications, including prescriptions of drugs, over-the-counter medications, supplements, and herbal remedies. It is a crucial component of a patient’s medical profile and is essential for ensuring safe, effective, and appropriate healthcare. The BPMH involves collecting information from various sources, such as the patient, his/her family members or caregivers, pharmacists, and medical records. It requires thorough communication between healthcare providers to avoid discrepancies and ensure the accuracy of the data. Clinical pharmacists can assist or play a key role, especially as part of interdisciplinary teams, in improving medication use, advising providers to respond to gaps in treatment/care needs, diminishing inappropriate prescribing practices, and improving therapeutics safety. Furthermore, an imprecise BPMH process could lead to an increased risk of errors during the medication reconciliation phase, resulting in an increase in Adverse Drug Reactions (ADRs) and Potentially Inappropriate Prescriptions (PIPs), which can determine patient’s unfavorable clinical outcomes, such as prolonged hospitalization time [3].

In Italy, due to the absence of an officially established clinical pharmacist’s role, medication review activities extend beyond the sole responsibility of pharmacists and are frequently delegated to various other healthcare professionals, including clinicians and nurses. When medication review activities are not delegated to pharmacists and are instead assigned to other healthcare professionals, several risks can arise. A key concern relates to the nuanced expertise required to assess complex drug interactions, make precise dosage adjustments, and identify potential adverse reactions quickly, all of which fall within the domain of clinical pharmacists. Delegating these responsibilities to less specialized personnel increases the likelihood that critical interactions or dosing errors will be missed, jeopardizing patient safety and treatment effectiveness. In addition, the lack of pharmacist involvement increases the potential for delayed recognition of subtle medication-related problems, such as drug-induced organ toxicity or pharmacokinetic subtleties, which may exacerbate adverse patient outcomes [2,3].

These primarily stem from the lack of specialized pharmaceutical expertise and comprehensive knowledge of medications. For these reasons, in 2014, the Italian Ministry of Health issued a distinct recommendation concerning the implementation of medication review activities to safeguard against inadequate prescriptions, emphasizing the indispensable role of pharmacists in every phase of the process [4]. However, despite this directive, the current state of affairs in Italy remains highly diverse, as certain hospitals autonomously advocate for clinical pharmacy activities, often focusing on vulnerable patients who face polypharmacy challenges [5].

While the surgical setting is not traditionally a primary area of intervention for clinical pharmacists, some studies have demonstrated their valuable contribution in this context. These studies highlight the positive impact of clinical pharmacists in mitigating medication omissions, addressing incompleteness in treatment plans and reducing the risk of adverse events associated with patient therapy. Incorporating clinical pharmacists into surgical teams can lead to improved medication management and enhanced patient safety during the perioperative period [3,6,7,8]. A 2.7-fold increase in the risk of experiencing postoperative complications has been revealed in surgical patients treated with preadmission medications compared to those not taking preadmission medications [9]. This underlines the importance of preventing unintentional interruptions to prescribed therapies unrelated to surgery during the preoperative period. Most of the patients undergoing major surgery are exposed to associated significant cardio-respiratory stress. Accordingly, any sudden or prolonged withdrawal of the current drug therapy in these patients could add significant risk to their surgery and complicate the outcome, especially considering that they are often 65 years old or older and potentially treated with an average of six preadmission medications. No research to date has evaluated the impact of preadmission medication errors in surgical patients in Italy.

Despite routine internal and external audits revealing that over 80% of patients in our university hospital have a BPMH documented within 24 h of admission by the ward staff, the accuracy of BPMH collection by pharmacists in a real-world scenario and the factors contributing to medication discrepancies remain poorly defined in the context of Italian healthcare. Furthermore, it is important to note that, despite the good percentage of BPMH documentation by the ward staff, there is a lack of qualitative data. Obtaining qualitative insights in this regard could provide valuable knowledge for improving medication reconciliation practices and minimizing medication discrepancies, ultimately leading to safer and more effective patient care.

The Padua University Hospital, a large tertiary referral facility with 1600 beds, offers a wide range of medical services, including emergency care, general medicine, oncology, cardiology, pneumology, general surgery, orthopedic surgery, cardiothoracic surgery, rehabilitation, geriatrics, mental health, and palliative care. Additionally, the hospital provides specialized services like heart, liver, and lung transplantation and comprehensive care for pediatric patients with acute, chronic, and rare diseases. Given the complexity of patients’ situations and their extensive medication needs, the pharmacy department plays a crucial role in overseeing medication management across all clinical and surgical areas of the hospital.

In this study, our objective was to explore the potential role of clinical pharmacists in the perioperative surgical ambulatory setting of our university hospital. Specifically, we aimed to assess the impact of the quality and accuracy of pharmacist-documented BPMHs during preoperative visits compared to the current gold standard represented by BPMHs documented by the ward staff, with a particular focus on identifying and addressing any omissions in medication information. By conducting this investigation, we sought to gain insights into how clinical pharmacists can contribute to optimizing medication reconciliation processes in the perioperative period, potentially enhancing patient safety and overall surgical outcomes.

## 2. Materials and Methods

### 2.1. Study Design

A prospective observational non-profit study was undertaken at major public university hospital in Padova, Italy, over a 20-week period (March to July 2021) and performed in three phases (see Figure 1):

i.During the first phase, which we refer to as the feasibility phase, our primary focus was to identify the specific and tailored information required to establish an accurate BPMH. This phase took place in March 2021 and involved the selection of 10 crucial items related to the patient’s therapy. To select the items to be included in the study, a multidisciplinary team consisting of a clinical pharmacist, ward staff (especially nurses), an anesthesiologist, and a surgeon who evaluated the information currently collected at the preoperative visit, considered the most relevant pharmacological information defined as “necessary”, and developed a list of 10 items. This list includes all the important information to be collected when a polypharmacy occurs, such as the brand name of the drugs and active ingredients prescribed, their pharmaceutical forms, doses (defined as a specific amount of medication taken at one time) and dosages (defined as how to take the medication as prescribed: a specific amount, number, and frequency of doses over a specific period of time), and all the information related to the initiation and duration of therapy. Additionally, we verify the completeness of information related not only to drugs but also to integrative therapies, homeopathic medicines, and dietary/herbal supplements, in order to achieve a complete BPMH collection. As our study aims to focus on the impact of the clinical pharmacist in the collection of the BPMH, we decided to focus on purely pharmacological items. Therefore, we did not include information such as comorbidities or diagnoses.ii.The second observational phase involved systematic data collection on the activity of ward staff (e.g., nurses) in gathering medication history during April–May 2021.iii.The third and last interventional phase examined the impact, measured by the omissions rate in this and previous phases, of BPMH collection carried out by the clinical pharmacist in optimizing the prescription appropriateness during June–July 2021.

During the latter two phases, all patients who had access to the preoperative outpatient clinic during the period analyzed were consecutively recruited into the study after signing formal informed consent. The consecutive research phases served a valuable purpose in assessing the rate of omissions in the BPMH collected by the nursing staff, which was considered current practice and was used as the control group. Subsequently, we compared these findings to those obtained by the clinical pharmacist in the intervention group.

In both cases, healthcare professionals gathered all relevant patient data during the preoperative anesthesiology visit. This process involved direct interviews with the patient and/or his/her caregiver, along with thorough review of online medical records and, when accessible, the general practitioner’s documentation of the patient’s clinical history. The final BPMH report was furnished to the anesthesiologist, serving as the foundational reference for informing them of any necessary therapeutic adaptations concerning the selection of anesthetic modalities and agents.

In this project, the clinical pharmacy service involved one dedicated hospital pharmacy resident, supervised by an attending clinical pharmacist. The role of the clinical pharmacist related with patient admissions comprised the following: acquiring a BPMH, performing the medication reconciliation, reviewing medication orders, pharmaceutical compounding, and managing medication supply. The clinical pharmacist reviewed the patients within 24 to 72 h of admission (e.g., patients hospitalized over the weekend were reviewed on Monday). If a PIP or discrepancy was identified, the clinical pharmacist acted by directly contacting the prescriber or specialist who would be managing the patient once hospitalized.

The primary outcome of the study was to measure the rate of total omissions in the two groups. These were observed and noted at the end of steps 2 and 3, respectively. In each of these, we checked the number of items not reported in each patient’s BPMH, according to the 10 items identified in step 1. The total amount of missing information, divided by the amount of information that should theoretically be collected and corresponding to what was reported in step 1, corresponded to the omission rate for each group.

Both the present manuscript and all parts of the study were checked and submitted according to the checklist for Strengthening the Reporting of Observational Studies in Epidemiology (STROBE) [10].

### 2.2. Ethics Approval

This study was conducted in accordance with Good Clinical Practice (GCP) recommendations, using the guidance documents and practices outlined by the International Conference on Harmonization and the European directives 2011/20/CE and ISO 4155, and it was in agreement with Italian regulations. All patients gave written informed consent to take part in the study, and the protocol was approved by the Padova Province Local Ethical Committee (authorization number: 5136/AO/21).

### 2.3. Inclusion and Exclusion Criteria

All patients with a scheduled surgery in 2021, admitted to the preoperative outpatient clinic of the General Surgery Department of the University Hospital of Padua, were recruited. Patients unable to sign a consent form or with fewer than three drugs in their BPMH were excluded from the analysis.

### 2.4. Statistical Analysis

The sample size of the study was calculated referring to the data already available in the literature on this topic [3]. G Power software (version 3.1.9.7) was used to better estimate the proportion of patients to include in the two groups [11]. The study was powered to 90% with 2-tailed significance α of 0.05. Assuming this, the number of patients to be recruited should have been at least 139, divided into the two groups. Continuous normally and non-normally distributed variables were reported as mean ± standard deviation (SD) or median (interquartile range, IQR), respectively, whereas dichotomous variables were expressed as frequencies and percentages. Statistical significance was calculated with an unpaired Student’s t-test for two-sample comparisons or the non-parametric Mann–Whitney’s U-test for independent variables, whereas categorical variables were compared using Pearson’s chi-square test or the Fisher’s exact test, using the R software [12]. The differences between groups were considered significant when *p* <0.05.

## 3. Results

In this comprehensive study, we enrolled a total of 140 patients, equally distributed into two groups: the intervention group and the control group.

Before initiating the study, we ensured that both groups were comparable in terms of baseline characteristics. This included factors such as median age, gender distribution, percentage of patients aged 65 or older, comorbidities, and polypharmacy, defined as the concomitant use of five or more medications per day. These efforts were taken to minimize any confounding variables and to establish a reliable basis for comparison between the groups.

However, upon conducting our analysis, we did identify statistically significant differences between the two groups in terms of the types of surgery performed, as indicated in Table 1. Despite these differences, we employed appropriate statistical methods to account for potential biases and to draw reliable conclusions from our findings.

Our investigation into the accuracy of BPMH yielded noteworthy results, with statistically significant differences observed for each of the ten items considered. Notably, the intervention group exhibited a remarkable proficiency in BPMH collection, with only 14 omissions (2.0%) detected, compared to a substantial 400 instances (57.4%) of missing information in the control group (*p* < 0.05) (Table 2).

## 4. Discussion

Many recent articles have discussed the critical importance of the pharmacist in surgical procedures, with a focus on the role in the perioperative multidisciplinary team and the impact of their collaboration on the outcome of surgical procedures [13]. As reported in the literature, an accurate medication collection process leads to more accurate reconciliation, reducing medication errors and adverse reactions due to incorrect medication prescription [14,15]. The introduction of the clinical pharmacist into the medication process has produced positive results in decreasing the amount of incomplete information and has been effective in making drug interviews more precise, consistent, and accurate. According to Nanji et al., 5.3% of medication administrations during 277 surgeries was incorrect, and 79.3% was avoidable [16]. 

While several studies have explored the potential role of clinical pharmacists in various settings, only a limited number have specifically investigated their involvement in presurgical settings, such as the study we conducted [17,18]. Notably, in the case of scheduled surgeries, establishing a meaningful interaction between the clinical pharmacist and the patient (or his/her caregivers) is crucial in order to achieve an effective BPMH. This is because transitions of care, especially during the perioperative period, are critical moments for patient safety, necessitating a meticulous medication reconciliation process and accurate information collection to prevent potential errors. 

Ensuring a complete and accurate documentation of a patient’s medications is of utmost importance for facilitating seamless communication among healthcare providers, particularly during the time of surgery. This documentation helps in clearly identifying which medications have been discontinued due to the surgical procedure or replaced over a specific period to address surgical risks. By involving clinical pharmacists in the presurgical process, healthcare facilities can enhance medication management, reduce the likelihood of medication-related complications, and ensure better patient outcomes during the perioperative period. 

The role of the pharmacist during surgical procedures is indisputably necessary for efficient healthcare delivery and positive patient outcomes. Indeed, the perioperative pharmacist has advanced therapeutic knowledge and experience to ensure the appropriate use of drugs and patient outcomes during the preoperative, intraoperative, and postoperative phases for all surgical patients [19]. During the preoperative period, the pharmacist ensures fluid status optimization, appropriate analgesia administration, antimicrobial prophylaxis, and venous thromboembolism prophylaxis. At the intraoperative stage, the pharmacist ensures that antimicrobial prophylaxis, fluid resuscitation, and anesthetic plans are redosed. Opioid doses are then adjusted based on the patient’s tolerance and adverse effects. During the postoperative period, the patient receives medication monitoring, withdrawal of strong opioids, and counseling [19]. Furthermore, the pharmacist is able to understand and participate in the analysis of the main causes of surgical complications: in fact, our study demonstrated the possibility of decreasing the risk of medication errors due to incorrect medication prescription or an inaccurate, incomplete, or absent medical history prior to hospitalization [20]. By meticulously examining the two groups, we gained crucial insights into the potential impact of involving clinical pharmacists in the BPMH collection process, as well as identifying any omissions that might occur in the existing approach led by nursing staff. In addition to a reduction in drug therapy errors, in fact, a higher percentage of patients in the intervention arm obtained accurate information on how to use the pharmaceutical form, the homeopathic medicine, or other dietary supplements. In the control arm, on the other hand, more than 400 omissions were recorded, most of which related to the active ingredient taken. The most surprising finding emerged when examining data related to treatment duration and initiation, along with the need for manipulation of medicinal products before oral administration. Astonishingly, none of these three aspects were ever considered or recorded for any patient within the control group. This difference in terms of omissions could be attributed to various reasons. These include information paucity of patient medication history, inaccuracies of primary care physicians’ referral letters, outdated drug history obtained from the general practitioner, and lack of time to collect a detailed medication history. Moreover, as far as the Italian situation is concerned, the inadequacy of training in pharmacology must also be taken into account, especially for health professionals other than pharmacists, where pharmacology and pharmacotherapy are taught in a single course lasting a few months [21]. Detecting medication omissions on time is of paramount importance to ensure patient safety and effective healthcare delivery. When crucial information about a patient’s medications is missing or incomplete, it can lead to potential medication errors, adverse reactions, or suboptimal treatment outcomes. Timely identification of these omissions allows healthcare providers, including clinical pharmacists, to promptly address the gaps in medication information and conduct a thorough medication review. By rectifying omissions in a timely manner, healthcare teams can make well-informed decisions, design appropriate treatment plans, and prevent any potential harm to the patient. Additionally, accurate and comprehensive medication documentation supports seamless communication among healthcare professionals, contributing to a more cohesive and integrated approach to patient care. Ultimately, early detection of medication omissions plays a crucial role in enhancing patient safety, improving medication management, and ensuring the best possible healthcare outcomes [22,23,24,25]. As demonstrated by Nguyen et al., the introduction of the clinical pharmacist in a perioperative environment determined an improvement in the BPMH, decreasing the average error rate from 5.25 to 0.21 errors per patient in the intervention arm (reconciliation performed by pharmacist) [3]. In addition, the introduction of a pharmacist in the surgical department helped to lower the amount of incompleteness by more than 96.0%.

Stratifying according to clinical setting, 27.0% of patients enrolled in the present study (19/70) had a tumor of the gastrointestinal tract, closely related to the scheduled surgery; the median age was 64 (IQR 48–80), and of these, 41.0% (11 patients) were taking more than five drugs per day. As shown by recent epidemiological studies, 50.0 percent of cancer patients over 60 years of age take more than five medications per day [26]. Polypharmacy is therefore a fairly common condition in these individuals and is a high-risk factor for ADRs, for the risk of interactions and for potentially inappropriate medication use. 

Furthermore, cancer patients are managed across the entire health care spectrum, both in acute hospital care and in home chemotherapy programs and other community settings. This complex system of interdependencies has a high potential for miscommunication, especially in changing hospital settings [27]. This is an area of interest for pharmacists, who possess the skills needed to optimize the transition of patient care across these settings by identifying errors and other medication-related problems [27]. The collaboration between pharmacists and surgeons has the potential to make a significant contribution, including improving the quality of care, containing costs for patients, and, above all, reducing mortality [28,29].

### 4.1. Strengths and Weaknesses (Study Limitations)

In the light of the findings obtained from our study, we are confident that the role of a pharmacist-led data collection service could be developed and implemented. As pioneered in an Australian hospital, a collaborative model between clinical pharmacists and physicians regarding data collection, medication reconciliation, and pharmacotherapy records management for in-patients was successfully introduced [24]. The service was positively received, and the pharmacists’ recording and reconciliation of medical records led to a statistically significant reduction in medication errors of over 80% versus the comparison arm without pharmacist intervention [22]. 

Based on the compelling evidence derived from our study, we firmly advocate for the implementation of a similar model in the broader context of Italian hospitals. This forward-looking approach entails expanding the role of pharmacists beyond their traditional focus on medication management. By actively fostering collaboration with multidisciplinary clinical teams, including those in the emergency department and on the wards, pharmacists can significantly fortify their presence in the clinical setting and make meaningful contributions to various aspects of patient care.

Embracing this collaborative model offers several advantages. First and foremost, it facilitates a seamless flow of information and expertise among healthcare professionals, enhancing communication and promoting a holistic approach to patient treatment. Pharmacists, with their specialized knowledge of medications, can actively engage in medication reconciliation, ensuring accurate and comprehensive medication histories for each patient. This proactive involvement plays a pivotal role in preventing medication errors and optimizing treatment plans, ultimately leading to improved patient safety and health outcomes.

Moreover, by integrating pharmacists into the healthcare team, a wealth of pharmaceutical knowledge and insights can be tapped into during patient consultations. Pharmacists can provide valuable recommendations, offer alternative medication options, and address any potential drug interactions, leading to more informed decisions regarding patient care.

The implementation of pharmacists within the multidisciplinary ward teams can significantly improve data collection and the key stages of medication reconciliation. This collaborative approach ensures that all relevant medication information is accurately gathered and reconciled, leading to higher quality patient care. By leveraging their expertise and working closely with other healthcare professionals, pharmacists can play a pivotal role in optimizing patient outcomes and streamlining the overall healthcare process.

Implementing such a model, as previously demonstrated via other examples, holds great potential for advancing patient care in Italian hospitals, allowing pharmacists to take on a more substantial and active role in the clinical setting. This proactive approach can lead to improved data management, enhanced medication reconciliation, and ultimately, a higher quality of care for patients [30,31].

While our present study has provided valuable insights, it is essential to acknowledge its limitations to inform future research and broaden the scope of investigation. One of the primary limitations is that the data collection was limited to a specific subset of surgical units, excluding other critical departments, such as emergency surgery. Moreover, the focus of our study was on the preoperative phase of the perioperative process. As a result, the impact of the clinical pharmacist in different perioperative settings remains unexplored and requires further investigation in a more comprehensive study.

Another limitation pertains to the focus on process endpoints rather than clinical outcomes. While we meticulously analyzed the amount of incomplete medication information, we did not evaluate efficacy or safety endpoints. Indeed, our study allows us to quantify the omissions observed in the two groups; however, it falls short of providing insight into the potential impact on patient well-being. This limitation arises from our exclusive focus on the preoperative phase, preventing us from tracing patients across the entirety of their perioperative trajectory and thus comprehending the broader implications of these omissions on their health outcomes. To gain a more holistic understanding of the clinical pharmacist’s impact, future analyses with a larger cohort will be imperative. These future investigations should aim to assess the proportion of patients who experienced one or more medication errors, the specific types of medication errors related to drug prescribing or administration, and the incidences and consequences of ADRs associated with medication errors. Additionally, we will explore factors such as hospitalization length, the number of drugs prescribed at hospital discharge, and the occurrence of drug interactions.

By expanding the scope of our research and incorporating clinical outcomes, we can delve deeper into the impact of the clinical pharmacist’s role in perioperative care. This comprehensive approach will enable us to identify potential areas of improvement and measure the true value of pharmacist-led interventions in enhancing patient safety and overall healthcare outcomes.

Furthermore, we have yet to gather feedback regarding the extent to which a comprehensive BPMH can enhance the workflow of the subsequent healthcare provider attending to the patient, specifically the anesthesiologist. Their insights are crucial for interpreting medication history in the context of anesthesia, potentially impacting anesthesia modality decisions, drug interactions, and patient safety during surgery.

In conclusion, while our current study has provided valuable groundwork, future research with a broader focus on diverse perioperative settings and clinical outcomes will offer a more comprehensive understanding of the clinical pharmacist’s role. Through ongoing investigations, we can make informed decisions to optimize medication management and elevate the quality of patient care within the Italian healthcare system.

### 4.2. Further Research

The insights gained from this analysis hold significant implications for planning the implementation of a comprehensive medication review process, including the critical reconciliation phase, within our specific clinical setting. By showcasing the valuable contribution of the clinical pharmacist as an integral part of a multidisciplinary healthcare team, we can lay the groundwork for optimizing patient care and medication management. However, the crucial future step, already foreseen in future studies, must include the evaluation of patient outcomes in order to better quantify the impact of the BPMH collection activity managed by the clinical pharmacist.

Moreover, the development of a new professional role for clinical pharmacists within the Italian National Health System presents an exciting opportunity to advance their practice in the field of hospital pharmacy. This potential expansion of their responsibilities and involvement in patient care can lead to a more fulfilling and impactful role for pharmacists, promoting their professional growth and expertise.

The collaboration among pharmacists, physicians, and nurses is central to this endeavor. By fostering strong teamwork and direct interaction with patients and their caregivers, this collaborative approach can enhance the effectiveness of the medication review process and ensure patient-centric care. Additionally, it has the potential to strengthen the role of the clinical pharmacist within the ward team, elevating their position in the healthcare landscape and contributing to an overall improvement in the quality of patient care.

Overall, the findings from this analysis have far-reaching implications, laying the foundation for a more integrated and patient-centered approach to medication management. Through this collaborative effort, the clinical pharmacist’s expertise can be fully harnessed, benefitting both healthcare professionals and patients alike. As we move forward, embracing this model of multidisciplinary collaboration holds great promise in shaping a more effective and efficient healthcare system, ultimately resulting in improved patient outcomes and enhanced quality of care.

## 5. Conclusions

The implementation of a novel organizational model, involving the inclusion of clinical pharmacists in the perioperative surgical setting, has shown significant advancements in BPMH collection compared to the previous standard of care, especially, as demonstrated, during the preoperative phase. This encouraging outcome highlights the valuable role of clinical pharmacists in optimizing medication management during the perioperative period.

However, to further enhance the impact and recognition of clinical pharmacists in the healthcare system, additional research is warranted. Focusing on aspects related to medication reconciliation and extending the service to other healthcare settings can provide valuable insights. Investigating the comprehensive medication management process beyond the entire perioperative setting will contribute to improving patient outcomes and the overall quality of care.

Promoting the visibility and recognition of clinical pharmacists in a country where their role is not yet fully established requires continued dedication and evidence-based research. By demonstrating the positive impact of clinical pharmacists on patient care and safety, healthcare institutions and policymakers can better appreciate and integrate this essential healthcare figure into various healthcare settings, ultimately enhancing the overall healthcare landscape.

## Figures and Tables

**Figure 1 pharmacy-11-00142-f001:**
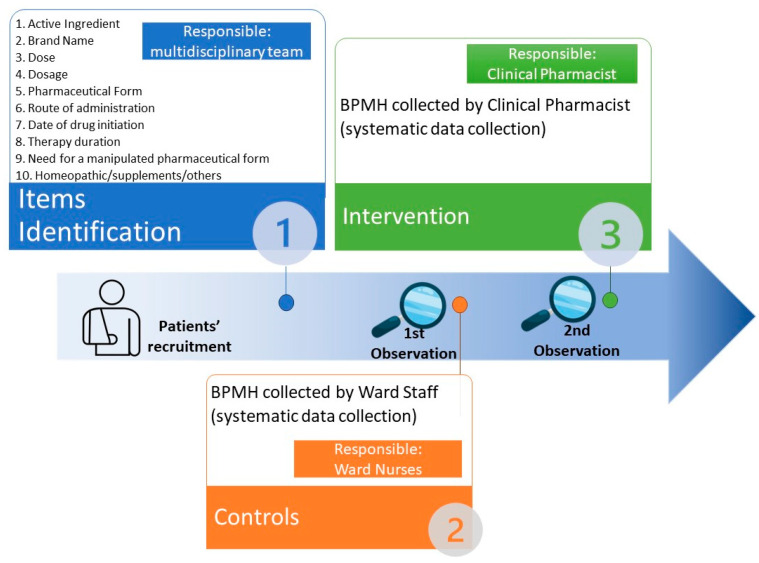
Graphical representation of the study phases.

**Table 1 pharmacy-11-00142-t001:** Baseline patient characteristics.

Variables	All Patients (*N* = 140)	Control Group (*N* = 70)	Intervention Group (*N* = 70)	*p*
Male, *n* (%)	75 (53.6)	37 (52.8)	38 (54.3)	0.86
Age, median (IQR)	61 (55–73)	60 (55–74)	64 (56–75)	0.76
Comorbidities, median (IQR)	3 (2–5)	3 (2.5–4)	3 (2–5)	0.80
Patients aged ≥ 65 years (%)	67 (47.8)	34 (48.6)	33 (47.1)	0.86
Number of medications taken daily per patient, median (IQR)	4 (3–5)	4(3–5)	4(3–6.0)	0.31
Patients with polypharmacy (drugs taken ≥5; %)	96 (68.6)	44 (62.8)	52 (74.3)	0.15
*Type of surgery*				<0.05
General surgery (%)	73 (52.1)	70 (100)	3 (4.3)	--
Gastric surgery (%)	62 (44.8)	0 (0.0)	62 (88.6)	--
Others (%)	5 (3.1)	0 (0.0)	5 (7.1)	--

**Table 2 pharmacy-11-00142-t002:** Comparative analysis of information accuracy regarding drug therapy of presurgical patients collected during the feasibility phase and the intervention period. (N.A.: not applicable).

Information Reported in the BPMH	Control Group (*N* = 70)	Intervention Group (*N* = 70)	*p*
Active pharmaceutical ingredient (%)	21 (30.0)	68 (97.1)	<0.05
Brand name (%)	53 (75.7)	69 (98.6)	<0.05
Route of administration (%)	44 (62.9)	70 (100)	<0.05
Pharmaceutical form (%)	53 (75.7)	70 (100)	<0.05
Dose (%)	55 (78.6)	68 (97.1)	<0.05
Dosage (%)	49 (70.0)	70 (100)	<0.05
Date of drug initiation (%)	0 (0.0)	64 (91.4)	N.A.
Drug therapy duration (%)	0 (0.0)	67 (95.7)	N.A.
Need for a manipulated pharmaceutical form (%)	0 (0.0)	70 (100)	N.A.
Homeopathic/supplements/others (%)	25 (35.7)	70 (100)	<0.05
Total omissions (%)	400/700 (57.1)	14/700 (2.0)	<0.05

## Data Availability

Data are available from the authors upon request to the corresponding author.

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
