# Peer review of "Best Possible Medication History Collection by Clinical Pharmacist in a Preoperative Setting: An Observational Prospective Study"

_pharmacy, 2023, doi:10.3390/pharmacy11050142_

Round 1
Reviewer 1 Report
Abstract
The content of the abstract includes all the information needed to summarize the manuscript. It requires some minor revisions:
-line 16: please correct “in an Italian perioperative”
-line 18: I suggest to modify the sentence “The study comprises 3 phases: feasibility, observational etc...”
-line 22: please uniform the number of significant digits (it should be 2.0%)
-line 25: please explain these percentages (also in the manuscript are not explained)
Introduction
The introduction provides sufficient background and it only requires some minor revisions:
-line 38: it should be “his/her medication” and not “their medication”
-line 44: it should be “his/her family members”. Please also consider change “pharmacy” with “pharmacists”
-line 50: please add “risk” (an increased risk of errors). Alternatively, it should be “an increase in errors”
-line 51: it should be “an increase in ADR”
-line 52: comma is missing before which
-line 96: please consider “pneumology” instead of “respiratory”
Methods
The study design seems appropriate but some clarifications are needed, such as how omissions were taken into account: if patients were interviewed about their pharmacological history, how can omissions be detected (or was electronic health data taken as a starting point for all patients)? Please clarify which kind of omissions were taken into account (line 156).
I also suggest to collect comorbidities for an effective and complete medication reconciliation (Figure 1, phase 1). Please justify why diagnosis were not collected. Also, in Figure 1 it is not clear if the dosage (point 5 of phase 1) is the prescribed dose or the dose of the branded medication. Consider to add a further point to phase 1 which could be any adverse reactions or other relevant clinical information experimented by patients.
Other minor revisions:
-line 120: it should be “was to identify”
-line 122: the number of drugs is not in Figure 1, phase 1, please adjust
-lines 128-130: this sentence is not clear, please rephrase
-line 131: it is not clear how the impact was measured, please explain
-line 140: it should be “his/her caregiver”
- lines 143-148 should be moved to the discussion section rather than the methodology section
-line 150: please move "in this project" at the start of the sentence
-line 152: the authors do not state how discrepancies and PIPs were handled, e.g. if the clinical pharmacist found a discrepancy/PIP in the patient's current medications, did he/she discussed with the clinician who modified the patient’s therapyaccordingly?
-line 180: comma is missing after respectively
-Figure 2, phase 2: pleas correct the word “carried” (one e)
Results
-line 188: how were the patients randomized? Please add this information in the method section
-lines 200-205 should be moved to the discussion section
-Table 1: please uniform the number of significant digits (also in the manuscript for all the percentages). Please also clarify the number of patients analysed: 96 patients (see Table 1) should be the population analysed according to the inclusion criteria in the method section (line 172), therefore the characteristics of the population analysed should be the characteristics of this population and not of the overall population (140). Please justify
-line 207: noteworthy results should be stated in the discussion section, not the result section
-lines 211-213 should be moved to the discussion section. Also, it is not clear what these percentages (line 213) represent
-lines 214-217: please consider moving to the discussion section
-Table 2: please explain why the total population is 140 and not 96 (96 patients with >/= 3 drugs, as stated in the method section) and how total omissions were calculated
Discussion and conclusions
These sections are extensive and adequate. Please correct some minor revisions:
-line 235: “his/her caregiver” and “to achieve”
-line 256: During with capital D
-line 289: please uniform the number of significant digits
-lines 337 and 346: it should be “medication reconciliation” and not “therapeutic reconciliation”
-line 361: ADR was already used, please correct
-please correct double numbering in the references
A few minor revisions are required (listed in the comments).
Author Response
- Abstract: The content of the abstract includes all the information needed to summarize the manuscript. It requires some minor revisions:
Thank you for your comments, we fixed all the issues you reported directly in the manuscript.
1.1 line 16: please correct “in an Italian perioperative”
Thank you, we corrected it.
1.2 line 18: I suggest to modify the sentence “The study comprises 3 phases: feasibility, observational etc...”
Thank you, we modified the sentence as you suggested.
1.3 line 22: please uniform the number of significant digits (it should be 2.0%)
Thank you, we uniformed the number with the above mentioned style (one digit for the decimal)
1.4 line 25: please explain these percentages (also in the manuscript are not explained)
Thank you for your kind observation. We added the following words after the percentage reported in line 25: “(98.0% of completed information for the intervention versus 42.6%; p<0.05)”.
2.Introduction
The introduction provides sufficient background and it only requires some minor revisions:
Thank you for your kind observation, we fixed all the issues you reported directly in the manuscript.
2.1 line 38: it should be “his/her medication” and not “their medication”
Thank you, we fixed it.
2.2 line 44: it should be “his/her family members”. Please also consider change “pharmacy” with “pharmacists”
Thank you, we accept your suggestion and we made the recommended changes in the text.
2.3 line 50: please add “risk” (an increased risk of errors). Alternatively, it should be “an increase in errors”
Thank you, we added the word “risk” as suggested.
2.4 line 51: it should be “an increase in ADR”
Thank you, we changed the sentence as you suggested.
2.5 line 52: comma is missing before which
Thank you, we fixed it.
2.6 line 96: please consider “pneumology” instead of “respiratory”
Thank you, we replaced “respiratory” with “pneumology”.
3.Methods
3.1 The study design seems appropriate but some clarifications are needed, such as how omissions were taken into account: if patients were interviewed about their pharmacological history, how can omissions be detected (or was electronic health data taken as a starting point for all patients)? Please clarify which kind of omissions were taken into account (line 156).
Thank you for your kind comment, which allows us to clarify the methods section. As our study is a prospective observational study, the omissions will be recorded after the intervention of the clinical pharmacist and the ward staff, respectively. The types of omissions recorded, in order to be standardized, fall within the list of 10 topics considered in phase 1 of the study. It was precisely this phase that made it possible to identify the main aspects of a patient's drug therapy to be considered in a BPMH, and for this reason we decided to consider omissions related to these 10 items. Indeed, in the subsequent phases, BPMHs were carried out and omissions were checked for each item analyzed in each patient.
We understand that this part needs to be clarified in the “Methods” and, for this reason, we added a sentence to better describe what we did (see line 179 of the revised paper): “These were observed and noted at the end of steps 2 and 3 respectively. In each of these, we checked the number of items not reported in each patient’s BPMH, according to the 10 items identified in step 1. The total amount of missing information, divided by the number of information that should theoretically be collected corresponding to what was reported in step 1, corresponds to the omission rate for each group”.
3.2 I also suggest to collect comorbidities for an effective and complete medication reconciliation (Figure 1, phase 1). Please justify why diagnosis were not collected. Also, in Figure 1 it is not clear if the dosage (point 5 of phase 1) is the prescribed dose or the dose of the branded medication. Consider to add a further point to phase 1 which could be any adverse reactions or other relevant clinical information experimented by patients.
Thank you for your suggestions. Unfortunately, at this point is not possible to add comorbidities to our Phase 1. We discussed the possibility of adding this item at the study design stage, but we preferred to focus on anything related to the pharmacological area, to try to demonstrate how the pharmacist's expertise in this area is indispensable in order to achieve the most comprehensive collection of information possible. We specified the reasons of our choice directly in the methods, as you suggested (see line 144 of the revised paper: “As our study aims to focus on the impact of the clinical pharmacist in the collection of BPMH, we decided to focus on purely pharmacological items. Therefore, we did not include information such as comorbidities or diagnoses.”).
Regarding your comment related to Figure 1, we corrected the image by inserting both "dosage" and "dose" and removing "posology". The latter appeared to be a typographical error resulting from an incorrect translation from Italian.
This is also a good opportunity for us to clarify what we mean by dose, which, according to the American Medical Association (AMA) Manual of Style, is a specific amount of medication taken at one time, and by dosage, which is how to take the medication as prescribed: a specific amount, number, and frequency of doses over a specific period of time. We added these two definitions directly in the text, precisely in the methods section, lines 140-141.
Regarding side effects or other clinical aspects, it is not possible to include them now since data collection has ended. We greatly appreciate your feedback which will be helpful to improve our daily practice and future research.
Other minor revisions:
3.3 line 120: it should be “was to identify”
Thank you, we fixed it.
3.4 line 122: the number of drugs is not in Figure 1, phase 1, please adjust
3.5 lines 128-130: this sentence is not clear, please rephrase
Thank you for your comment, we fixed the two points above mentioned rephrasing lines from 136 to 147. First of all, we would like to clarify that the number of drugs is not one of the 10 topics related to phase 1 because we consider it a mandatory information to collect during the BPMH. It is the result of the sum of the different drugs taken by the patient. Anyway, we reformulated the sentence in order to make it clearer.
“This list includes all the important information to be collected when a polypharmacy occurs, such as the brand name of the drugs and active ingredients prescribed, its pharmaceutical form, dose (defined as a specific amount of medication taken at one time) and dosage (defined as how to take the medication as prescribed: a specific amount, number, and frequency of doses over a specific period of time), and all the information related to the initiation and duration of therapy. Additionally, we verify the completeness of information related not only to drugs but also to integrative therapies, homeopathic medicines, and dietary/herbal supplements, in order to achieve a complete BPMH collection As our study aims to focus on the impact of the clinical pharmacist in the collection of BPMH, we decided to focus on purely pharmacological items. Therefore, we did not include information such as comorbidities or diagnoses.”.
3.6 line 131: it is not clear how the impact was measured, please explain
Thank you for your comment. We added a statement to better explain the concept we meant. The rephrase sentence is the following (the added part is represented by the underlined words): “The third interventional phase examined the impact, measured by the omissions’ rate in this and previous phases, of BPMH collection carried out by the clinical pharmacist in optimizing the prescription appropriateness during June-July 2021”.
3.7 line 140: it should be “his/her caregiver”
Thank you, we fixed it.
3.8 lines 143-148 should be moved to the discussion section rather than the methodology section
Thank you for your kind suggestion. We moved these sentences to the discussion.
3.9 line 150: please move "in this project" at the start of the sentence
Thank you, we fixed the sentence as suggested.
3.10 line 152: the authors do not state how discrepancies and PIPs were handled, e.g. if the clinical pharmacist found a discrepancy/PIP in the patient's current medications, did he/she discussed with the clinician who modified the patient’s therapy accordingly?
Thank you for your comment. We added a sentence at the end of that line to better explain how we handle a PIP.
3.11 line 180: comma is missing after respectively
Thank you, we fixed the sentence as suggested.
3.12 Figure 2, phase 2: pleas correct the word “carried” (one e)
Thank you, we corrected the figure.
4.Results
4.1 line 188: how were the patients randomized? Please add this information in the method section
Thank you for your comment. However, the patients were not randomized. As ours was a prospective observational study, data were collected from consecutive patients who accessed our preoperative outpatient clinic. As you suggested, we added a sentence in the methods, directly after the presentation of the three phases: “During the latter two phases, all patients who had access to the preoperative outpatient clinic during the period analyzed were consecutively recruited into the study after signing formal informed consent.”
4.2 lines 200-205 should be moved to the discussion section
Thank you for your comments. We agree with the possibility of moving it to the discussion section, but after re-reading the text thanks to your suggestions, we feel that this part of the text is redundant in both the results section and the discussion section. For this reason, we preferred to remove it from the manuscript to make it smoother and more readable.
4.3 Table 1: please uniform the number of significant digits (also in the manuscript for all the percentages). Please also clarify the number of patients analysed: 96 patients (see Table 1) should be the population analysed according to the inclusion criteria in the method section (line 172), therefore the characteristics of the population analysed should be the characteristics of this population and not of the overall population (140). Please justify
Thank you for your suggestions. We uniformed the significant digits in both the tables and manuscript. Regarding the number of patients, there was a typo in Table 1. 96 are the patients on polypharmacotherapy (more than 5 drugs per day). 140 are the patients who met the inclusion criteria and were included in the study. We have corrected the reference in Table 1. Thank you again for pointing out the error.
4.4 line 207: noteworthy results should be stated in the discussion section, not the result section
Thank you for the suggestion. The presence of this sentence in the results section, in our opinion, makes the text more fluid and ensures better readability for the reader. It is just a quick presentation of a result that is then commented on in the discussion. For this reason, we think it might be useful to keep this small sentence in the results and hope that you can accept this decision of ours.
4.5 lines 211-213 should be moved to the discussion section. Also, it is not clear what these percentages (line 213) represent
Thank you for your suggestion. We decided to We decided to remove this sentence as it is likely to confuse the reader. For information purposes only, we point out that the percentages reported are nothing more than the reciprocal of the rate of omissions detected by group.
4.6 lines 214-217: please consider moving to the discussion section
Thank you for your suggestion. We moved this part in the discussion.
4.7 Table 2: please explain why the total population is 140 and not 96 (96 patients with >/= 3 drugs, as stated in the method section) and how total omissions were calculated
See the answer to your suggestion nr. 4.3
5.Discussion and conclusions
These sections are extensive and adequate. Please correct some minor revisions:
5.1 line 235: “his/her caregiver” and “to achieve”
Thank you for your suggestions. We fixed them.
5.2 line 256: During with capital D
Thank you for your suggestion. We fixed it.
5.3 line 289: please uniform the number of significant digits
Thank you for your suggestion. We fixed it.
5.4 lines 337 and 346: it should be “medication reconciliation” and not “therapeutic reconciliation”
Thank you for your suggestion. We corrected the two errors.
5.5 line 361: ADR was already used, please correct
Thank you for your suggestion. We fixed it.
5.6 please correct double numbering in the references
Thank you for your comment. We corrected them.
Reviewer 2 Report
It was said: When medication review activities are not delegated to pharmacists and areinstead assigned to other healthcare professionals, several risks can arise. The content here can be enriched, such as which risks. Also, the research flowchart is too vague, and the ten projects in the first stage of the research.
Anesthesiologists should have a significant impact on perioperative medication, especially anesthetics, which seems to be not clearly explained in the research design and discussion sections.
There are flaws in the research design. If insufficient consideration is given to the impact of anesthesia methods, whether the data collector collects data before talking to the anesthesiologist, etc. on the results. These differences may be factors that affect the patient's medication history,
The perioperative period includes preoperative, intraoperative, and postoperative periods. Can the topic be clearly defined as the preoperative preparation stage?
It is ok.
Author Response
We really appreciated Reviewer 2 comments and suggestions and we hope we satisfied all of them. Please, see below the detailed point-by-point response.
1.It was said: When medication review activities are not delegated to pharmacists and are instead assigned to other healthcare professionals, several risks can arise. The content here can be enriched, such as which risks. Also, the research flowchart is too vague, and the ten projects in the first stage of the research.
Thank you for your precise consideration. We have expanded the concept by inserting a specific sentence that better specifies how the specific training of the clinical pharmacist can be useful in this context. The added sentences are the following: “A key concern relates to the nuanced expertise required to assess complex drug interactions, make precise dosage adjustments, and identify potential adverse reactions quickly, all of which fall within the domain of clinical pharmacists. Delegating these responsibilities to less specialized personnel increases the likelihood that critical interactions or dosing errors will be missed, jeopardizing patient safety and treatment effectiveness. In addition, the lack of pharmacist involvement increases the potential for delayed recognition of subtle medication-related problems, such as drug-induced organ toxicity or pharmacokinetic subtleties, which may exacerbate adverse patient outcomes.”.
Regarding the comment on the flowchart showing the study design, it is only a graphical representation of the three phases of the study. In any case, we have tried to improve the quality and content of the figure by inserting the persons responsible for each phase (clinical pharmacist, staff), correcting typos (e.g. "posology" is a typo in a translation from Italian), and inserting the timing of patient recruitment and the execution of the different phases.
The 10 items chosen in phase 1 were discussed by a multidisciplinary team of professionals (clinical pharmacist, ward staff, surgeon, anesthesiologist) as 10 pharmacological information to be collected preoperatively. We added this aspect in the methods part. This is the added sentence: “To select the items to be included in the study, a multidisciplinary team consisting of clinical pharmacist, ward staff (especially nurses), anesthesiologist, and surgeon evaluated the information currently collected at the preoperative visit, considered the most relevant pharmacological information defined as "necessary," and developed a list of 10 items.”
2.Anesthesiologists should have a significant impact on perioperative medication, especially anesthetics, which seems to be not clearly explained in the research design and discussion sections.
Thank you for your comment. The role of the anesthesiologists is unquestionable. So much so that they are the beneficiaries of a well-collected BPMH, using the information collected by the clinical pharmacist to better manage anaesthesia. As your comment #4 rightly points out, our study mainly investigates the effectiveness of the BPMH collected by the pharmacist in the preoperative phase. Therefore, the type of anaesthesia and the anaesthetic agents used subsequently were not taken into account. We, as clinical pharmacists, had the opportunity to see the patient before the anaesthetic examination and, according to Italian law, we could only return to the anesthesiologist a report of the drug therapy taken by the patient (more precisely, the BPMH). The responsibility for suspending, modifying or substituting certain drugs remains a prescription and therefore the sole responsibility of the medical doctor.
To be more precise in the paper, we have corrected the timing of our intervention: no longer in the entire perioperative phase, but in the preoperative phase. Furthermore, as already reported in the response to your first comment, we emphasized that the item selection activity (phase 1) was carried out in agreement with a multidisciplinary team in which the anesthesiologist was present.
Moreover, we reported that the results of phases 2 and 3 (a BPMH report) was given to anesthesiologists to help them choose the most appropriate anesthetic methods and anesthetic drugs. In addition, the BPMH served as a basis for the anesthesiologist to recommend to the patient any medication to be withheld for the surgery. We have added a few sentences to the Methods section which, we hope, will help resolve your observation, better specifying the role of the anesthesiologist and, above all, identifying the moment of the BPMH as prior to the anesthesiologic visit.
We also added a sentence in the discussion (subsection “4.1 Strengths and Weaknesses”) in which we in which we report on how the results of the BPMH should be shared with anesthesiologists to see if there is a need for improvement or special focus. The sentence is the following: “Furthermore, we have yet to gather feedback regarding the extent to which a comprehensive BPMH can enhance the workflow of the subsequent healthcare provider attending to the patient, specifically the anesthesiologist. Their insights are crucial for interpreting the medication history in the context of anesthesia, potentially impacting anesthesia modality decisions, drug interactions, and patient safety during surgery.”
3.There are flaws in the research design. If insufficient consideration is given to the impact of anesthesia methods, whether the data collector collects data before talking to the anesthesiologist, etc. on the results. These differences may be factors that affect the patient's medication history.
Thank you for your kind suggestions. As reported in the previous two points, the aim of our study was to verify the impact of the collection of BPMH by the clinical pharmacist during the preoperative phase, in particular at a time prior to the anaesthetic visit. Furthermore, BPMH, as defined by the American College of Clinical Pharmacy, consists of collecting information on the patient's current drug therapy, without taking into account the drugs that the patient will have to take later (e.g. anaesthetics). For this reason, information on anaesthesia was not collected. We have explained in the previous points and in the text how the anaesthesiological visit proceeds following the moment of collecting the BPMH (subject of the present study). For this reason, we have specified in several points that our intervention took place in the preoperative phase. Also, accepting your suggestion to extend the relationship with the anesthesiologists, we have better explained where this was involved, with what potential benefits, with what limitations, and how it is necessary to get feedback from the anesthesiologist him/herself on the quality of a BPMH collected by a clinical pharmacist compared to normal clinical practice.
4.The perioperative period includes preoperative, intraoperative, and postoperative periods. Can the topic be clearly defined as the preoperative preparation stage?
Thank you for your comment. As previously explained, we accepted your suggestion and we re-modulated the focus of our paper on the preoperative phase, as the initial step of the entire perioperative process. We implemented the Methods section in order to explicit this concept.
Reviewer 3 Report
Thank you for the opportunity to review this study on the role of pharmacists in the improvement the medication review process, ´by improving the collection of medication history.
I have some comments:
1. The term "elderly" is considered ageist and should be avoided.
2. Improve Figure 1, which is hard to read.
- In line 177, the authors affirm their use the G power to estimate the proportion of patients to include into the two groups. Could you provide the power of the sample?
4. Please define polypharmacy and polypharmacotherapy.
5. The medication history for the control group was collected by health professional, please can you explain why they omit so much information.
- Could you discuss the relevance of the missing information in the control group, focusing on the type of information that is missing, not only in the number or percentage but also focusing on the practical utility of the information that is lacking?
Author Response
We really appreciated Reviewer 3 comments and suggestions and we hope we satisfied all of them. Please, see below the detailed point-by-point response.
Thank you for the opportunity to review this study on the role of pharmacists in the improvement the medication review process, ´by improving the collection of medication history.
I have some comments:
1.The term "elderly" is considered ageist and should be avoided.
Thank you for your kind suggestion. We have replaced the word "elderly" with the actual age taken into consideration (eg patients aged 65 or older).
2.Improve Figure 1, which is hard to read.
Thank you very much for your comment. The figure inserted in the manuscript is of lower quality than the one we have attached, as a separate file, to the submission. In any case, we have tried to improve the image, also in agreement with the comments of other reviewers, to make it easier to read and more impactful for the reader.
3.In line 177, the authors affirm their use the G power to estimate the proportion of patients to include into the two groups. Could you provide the power of the sample?
Thank you for your comment. We added a sentence, to the Statistical Analysis section, to better explain what we achieve via G power: “The study was powered to 90% with 2-tailed significance α of .05. Assuming this, the number of patients to be recruited should have been at least 139, divided into the two groups”.
- Please define polypharmacy and polypharmacotherapy.
Thank you for your suggestion that gave us the opportunity to better explain what we intend about polypharmacy and polypharmacotherapy. Although there is much debate in the literature about the definition of polypharmacy, we consider it to be the daily use of 5 or more drugs. We have included a sentence on this in the manuscript. In our intention, polypharmacotherapy is a synonym for polypharmacy. However, to avoid misunderstandings, we have decided to remove polypharmacotherapy from the manuscript and replace it with the term polypharmacy. We have also changed the reporting of polypharmacy and polypharmacotherapy in Table 1.
5.The medication history for the control group was collected by health professional, please can you explain why they omit so much information.
We appreciate the reviewer's feedback. Indeed, this is something we have also considered. The main reason, which is also supported by the literature, could be related to the lack of adequate training in the pharmacological field. In fact, in Italy, university courses for health professionals (excluding pharmacists) often include at most one course in pharmacology and pharmacotherapy for the entire duration of the course. This probably results in an incomplete basic knowledge of pharmacology, perhaps based on practical experience and therefore partial. In addition, the lack of an "official" clinical pharmacist in Italian hospitals means that there is little interest in the subject. If we add to this the permanent problem of the shortage of health personnel in many realities of our country, we can understand why this activity is not carried out in the best way. We have added a sentence to the discussion: “Moreover, as far as the Italian situation is concerned, the inadequacy of training in pharmacology must also be taken into account, especially for health professionals other than pharmacists, where pharmacology and pharmacotherapy are taught in a single course lasting a few months.”
6.Could you discuss the relevance of the missing information in the control group, focusing on the type of information that is missing, not only in the number or percentage but also focusing on the practical utility of the information that is lacking?
We appreciate the reviewer's feedback. Unfortunately, we do not have any real feedback on what the omissions did to the patients' health because our study was completed before the actual surgery and admission of the patients. We can only speculate about what the omissions might mean for patient outcomes. In this respect, we have added a few sentences to the discussion in order to better contextualize the matter. In particular, in the ' Strengths and Weaknesses' section, we have explained how this lack of information is a limitation of our study (“Indeed, our study allows us to quantify the omissions observed in the two groups; however, it falls short of providing insight into the potential impact on patient well-being. This limitation arises from our focus on the preoperative phase exclusively, preventing us from tracing patients across the entirety of their perioperative trajectory and thus comprehending the broader implications of these omissions on their health outcomes.”. On the other hand, in the 'Further research' section, we have reported how clinical outcomes will be the subject of future studies: “However, the crucial future step, already foreseen in future studies, must include the evaluation of patient outcomes in order to better quantify the impact of the BPMH collection activity managed by the clinical pharmacist”.